

# A forecast model for prevention of foodborne outbreaks of non-typhoidal salmonellosis

Fernando Rojas and Claudia Ibacache-Quiroga

Centro de Micro-Bio Innovación, Universidad de Valparaíso, Valparaíso, Chile
Escuela de Nutrición y Dietética, Facultad de Farmacia, Universidad de Valparaíso, Valparaíso, Chile

## ABSTRACT

**Background**. This work presents a forecast model for non-typhoidal salmonellosis outbreaks.

**Method**. This forecast model is based on fitted values of multivariate regression time series that consider diagnosis and estimation of different parameters, through a very flexible statistical treatment called generalized auto-regressive and moving average models (GSARIMA).

**Results**. The forecast model was validated by analyzing the cases of *Salmonella enterica* serovar Enteritidis in Sydney Australia (2014–2016), the environmental conditions and the consumption of high-risk food as predictive variables.

**Conclusions**. The prediction of cases of *Salmonella enterica* serovar Enteritidis infections are included in a forecast model based on fitted values of time series modeled by GSARIMA, for an early alert of future outbreaks caused by this pathogen, and associated to high-risk food. In this context, the decision makers in the epidemiology field can led to preventive actions using the proposed model.

## INTRODUCTION AND BIBLIOGRAPHICAL REVIEW

Non-typhoidal salmonellosis is a foodborne illness considered as a major health issue with a great impact on the economy and the food industry. This diarrhea-producing pathology is globally distributed, with 93 million cases worldwide and 155,000 deaths per year (*Majowicz et al., 2010*). The main challenges for the control of this pathology are the environmental ubiquity of the pathogenic agents, their spreading pathways and their presence in food (*Hamlet et al., 2006*). *Yerushalmy & Palmer (1959)* proposed the term epidemetrics, which emphasized the quantitative nature of epidemiologic studies. In this scenario, new models for the accurate prediction of infectious outbreaks are needed. *Brauer, Castillo-Chavez & Feng (2019)* focused on the development of these models and highlighted the need to update them according to the advances in statistical science. Currently, a new generation of surveillance strategies have emerged, whose help to detect emerging infections and identify a high risk of outbreaks of infectious diseases, related to climate change and other factors. Traditional surveillance methods are based on retrospective strategies, therefore the development of new epidemiological models that allow the prediction of

Corresponding author
Fernando Rojas, fernando.rojas@uv.cl

infectious outbreaks is of great interest, in order to take the necessary measures to limit their expansion and impact on public health (*Rees et al., 2019*). In these types of methods, risk factors (explanatory variables, such as open source internet data) are used to predict the outcome of interest (for example, number of cases reported) (*Ginsberg et al., 2009*). For example, *Santillana et al. (2014)* used these types of regression models to forecast the number of cases of seasonal influenza in the future. In this context, regression methods can be expanded using machine learning algorithms, to find complex associations between the result and the explanatory variables (*Santillana et al., 2015*).

Many of the current epidemetric models, like simulation, were implemented several years ago and need to be updated (*Mori, 1996*). In this scenario, the control of non-typhoidal salmonellosis should include new innovative surveillance and forecast statistical tools for its prevention, specially focused on the production and consumption of high-risk food (*Boyen et al., 2008*; *Ashton et al., 2016*) and improving the existing predicting tools (*Thakur & Anbanandam, 2017*). Non-typhoid salmonellosis outbreaks have been widely associated to climatic and environmental factors, like extreme rainfall, flooding, increased average temperatures, augmented frequency of extreme high-temperatures and changes in weather seasonal patterns (*Amuakwa-Mensah, Marbuah & Mubanga, 2017*). In this context, these parameters should be considered for the development of new models.

One of the strategies used to evaluate the link between number of infections and predictive variables (co-variables) for the monitoring, prediction and measurement of the impact of interventions is the analysis of time series, nevertheless, the vast majority of the approximations using Gaussian methods are prone to inaccuracies when the case counts are low and the bias of the statistical distribution of the real data is not symmetric (*Allen, 2017*). Therefore, appropriate statistical methods are required for count data, which allows the identification of infectious outbreaks through high precision forecasting.

Often, the count of reports of infections is a random variable that depends on time (*Konradsen et al., 2000*). In order to adequately describe this variable, the modeling of the possible temporal dependence related to multiple periods is needed. This time dependence of a variable can be described by an autoregressive moving average (ARMA) model in a time series.

The advantage of considering an ARMA framework lies in its malleability to model time-dependent variables, its easy estimation and interpretation, and its prediction power (*Gilbert, 2005*). On the other hand, its main disadvantages are related to its linear formulation between the response indexed in the time and its predictors, like the assumption that the modeled variable follows a Gaussian probability distribution. To overcome this last restriction, the original data are transformed, generating a complication in the interpretation of results (*Fischer & Kamps, 2011*).

To improve these limitations, *McCullagh & Nelder (1983)* formulated generalized linear models (GLM). In these models, the error is not modeled like in the classic regression models, but it is assumed that the response variable can be modeled by some type of statistical distribution belonging to the exponential family, where the normal distribution is a particular case. This approach allows linear and non-linear linking of the mean of the statistical distribution that best fits the real data of the response variable,

respect the predictors through a linking function. In this context, *Benjamin, Rigby & Stasinopoulos (2003)* proposed a GLM version of ARMA models known as Generalized ARMA (GLARMA). In this model, the systematic component allows us to express a function of the mean (the link function) by means of an additive arrangement, with parameters or coefficients that indicate the direction and magnitude of the relationship with explanatory variables (covariates) and the autoregressive and moving average components. This type of formulation gains in flexibility and the possibility of using other types of non-linear associations, under the ARMA framework. The parameters of a GLARMA model can be estimated using the maximum likelihood (ML) method, assuming a statistical distribution of the exponential family for the response variable. Often, a normal (or Gaussian) distribution is considered in the modeling of random variables (*Box et al., 2015*), but other distributions might also be assumed (*Rojas, 2016*; *Rojas et al., 2019*). ARMA and GLARMA models are often used to predict future values (*Benjamin, Rigby & Stasinopoulos, 2003*; *Calfa, 2015*). GLARMA models are also used to estimate mean values and find the conditional probability density function (PDF) to past data, like what occurs with random variables when temporal dependence and covariates are present. This last aspect is of particular interest in stochastic predictive models.

If the time series for modeling is non stationary and/or a stochastic seasonal component is considered, the GLARMA model described by *Benjamin, Rigby & Stasinopoulos (2003)* is appropriate. To overcome the aforementioned restrictions, GLARMA evolved into the generalized multiplicative seasonal autoregressive integrated moving average (GSARIMA) models, considering the differentiation and seasonality components in its formulation, see *Briët, Amerasinghe & Vounatsou (2013)*.

The aim of this paper is to propose a forecast model based on fitted values of multivariate time series and its projection, for prevention of non-typhoidal salmonellosis outbreaks, considering diagnosis and estimation of parameters obtained from GSARIMA model, given the high predicted accuracy that this model can achieve, which makes it a useful tool for epidemiological prevention.

## METHODOLOGY

### GLM, GLARMA and GSARIMA framework

Let $Y$ be an RV related to the counting reports of infections by a pathogen of interest such as *Salmonella enterica*. We consider that statistical distribution of $Y$ it belongs to the exponential family and that its PDF is

$$f_Y(y; \vartheta, \varphi) = \exp\left(\frac{y\vartheta - b(\vartheta)}{\varphi} + c(y, \varphi)\right), \quad y \in R_Y, \tag{1}$$

where $\vartheta, \varphi$ are canonical and scale parameters, respectively, $R_Y$ is the support of $Y$ and $b, c$ are specific functions whose characterize a member of the exponential family of statistical distributions.

This parametrization has the following properties for the description of mean and variance as the first and second derivatives of the canonical and scale parameters, where $E(Y) = b'(\vartheta)$ and $Var(Y) = \varphi b''(\vartheta)$, respectively.

In GLM, a function denominated link function of the mean of $Y$ is equal to a systemic component, then $g(\mu) = \eta$. In turn, the mean of $Y$ corresponds to the inverse link function of $\eta$, which is related to know values of $r$ covariates $\boldsymbol{x} = (\boldsymbol{x}_0, \boldsymbol{x}_1, \ldots, \boldsymbol{x}_r)^\top$, with $\boldsymbol{x}_0 = 1$, where

$$\mu = \mathrm{E}(Y) = g^{-1}(\eta) = g^{-1}(\boldsymbol{x}^\top \boldsymbol{\beta}), \tag{2}$$

with $\boldsymbol{\beta} = (\beta_0, \beta_1, \ldots, \beta_r)^\top$ being the regressors associated with $\boldsymbol{x}$.

Now we will derive the expressions for the case of an RV as $Y$ but ordered in a temporal sequence $t$, this RV indexed over time $t$, with $t = 1, \ldots, n$ is denominated $Y_t$. The conditional distribution of $Y_t$ given the past data set

$$\boldsymbol{H}_t = \{\boldsymbol{x}_1, \ldots, \boldsymbol{x}_t, y_1, \ldots, y_{t-1}\}, \tag{3}$$

which is also assumed to belong to the exponential family of statistical distributions. In this context, the past data conditional PDF similar to Eq. (1) is expressed as

$$f_{Y_t|\mathbf{H}_t}(y_t; \vartheta_t, \varphi) = \exp\left(\frac{y_t \vartheta_t - b(\vartheta_t)}{\varphi} + c(y_t, \varphi)\right), \quad y_t \in R_{Y_t}.$$

Note that here the canonical parameter $\vartheta_t$ and values of covariates $\boldsymbol{x}_t$ depend on a temporal sequence, while the parameter $\varphi$, remains constant and independent of the time sequence. In this conditions, we denote the conditional mean and variance of $Y_t$ given $\mathbf{H}_t$ by $\mu_t = \mathrm{E}(Y_t|\mathbf{H}_t) = b'(\vartheta_t)$ and $\mathrm{Var}(Y_t|\mathbf{H}_t) = \varphi b''(\vartheta_t)$, for $t = 1, \ldots, n$, respectively. $g(\mu_t)$ can be expressed as a GLARMA model of $p$ and $q$ orders. This model is denoted by GLARMA$(p, q)$, where

$$\eta_t = g(\mu_t) = \boldsymbol{x}_t^\top \boldsymbol{\beta} + \sum_{h=1}^{p} \phi_h(g(y_{t-h}) - \boldsymbol{x}_{t-h}^\top \boldsymbol{\beta}) + \sum_{j=1}^{q} \lambda_j(g(y_{t-j}) - \eta_{t-j}). \tag{4}$$

In this model $\phi_h$ and $\lambda_j$ correspond to the $h$th and $j$th components of an ARMA$(p, q)$ model, related to the autoregressive and moving average components, respectively. $\boldsymbol{\beta}$ are the regressors, related with known values of $r$ covariates, depending over time, denoted by $\boldsymbol{x}_t = (x_0, x_{1t}, \ldots, x_{rt})^\top$, with $x_0 = 1$. The link function $\eta_t = g(\mu_t)$ of the GLARMA model given in Eq. (4) can any as the identity function inverse or logarithmic (log) functions (allowing to consider non-linear associations). In this model the variance is assumed to be constant over time. In the case of the identity link function, we have that

$$\mu_t = \boldsymbol{x}_t^\top \boldsymbol{\beta} + \sum_{h=1}^{p} \phi_j(y_{t-h} - \boldsymbol{x}_{t-h}^\top \boldsymbol{\beta}) + \sum_{j=1}^{q} \lambda_j(y_{t-j} - \mu_{t-j}). \tag{5}$$

The above models can be extended to GSARIMA $(p, d, q) \times (P, D, Q)_s$ analogues by including seasonality ($S$) and differentiation ($D$) components as follows:

$$g(\mu_t) = \Phi(L)(1-L)^d(1-L^s)^D \Phi^P(L)(\boldsymbol{x}_t^\top \boldsymbol{\beta} - g(y_t)) + g(y_t) - \Lambda^Q(L)\Lambda(L^s)(g(y_t)$$
$$- g(\mu_t)) + g(y_t) - g(\mu_t),$$

where $s$ is the length of the period ($s = 12$ for monthly data with an annual cycle), $\Phi^P(L^s) = 1 - \phi_1^P L^s - \cdots - \phi_p^P L^{s^P}$, $\Lambda(L^s)^Q = 1 - \lambda_1^Q L^s - \cdots - \phi_q^Q L^{s^Q}$.

*Parameter estimation*

Considering $n$ realizations of $Y_t$, for $t = 1, \ldots, n$, $y_1, \ldots, y_n$, the likelihood function corresponds to the product of multiple conditional PDFs of $Y_t$ given the past observations $\mathbf{H}_t$. In this context, considering $\boldsymbol{\theta} = (\boldsymbol{\beta}^\top, \vartheta_t, \varphi, \boldsymbol{\phi}^\top, \boldsymbol{\lambda}^\top)^\top$ as the vector of model parameters to be estimated, the associated log-likelihood function for $\boldsymbol{\theta}$ is expressed by

$$\ell(\boldsymbol{\theta}) = \sum_{t=1}^{n} \log\left(f_{Y_t|\mathbf{H}_t}(y_t; \boldsymbol{\theta})\right). \tag{6}$$

The ML estimate of $\boldsymbol{\theta}$, $\widehat{\boldsymbol{\theta}}$, are obtained from the derivation of Eq. (6) respect each parameter $\boldsymbol{\beta}$, $\vartheta_t$, $\varphi$, $\boldsymbol{\phi}$ and $\boldsymbol{\lambda}$. The statistical inference of $\boldsymbol{\theta}$ is based on the asymptotic normality of the ML estimator $\widehat{\boldsymbol{\theta}}$.

*Model checking and diagnostic*

The random quantile (RQ) residual is used to diagnose the adequacy of the GLSARMA or GSARMA models to the data. RQ residual is mathematically defined as $r_t$ and can be calculated by:

$$r_t = \Phi^{-1}(F_{Y_t|\mathbf{H}_t}(y_t; \widehat{\boldsymbol{\theta}})), \tag{7}$$

where $F_{Y_t|\mathbf{H}_t}$ is the cumulative distribution function (CDF) of $Y_t$ conditional to past data, $\widehat{\boldsymbol{\theta}}$ is the ML estimate of $\boldsymbol{\theta}$, and $\Phi^{-1}$ is the inverse CDF of a standard normal distribution. RQ residual follows the standard normal distribution. For more details about this residual, see *Dunn & Smyth (1996)*.

To properly define a GSARIMA or GLARMA model, each competing model must be compared based on different combinations of order $p, q$. The global deviation (GD) is used as an indicator to compare these models. GD is $-2$ times the logarithmic probability ratio of the reduced model (in our case a GLM) and the complete model (in our case the GLARMA or GSARIMA model). To select the best competing model, the Akaike (AIC) and Bayesian (BIC) information criteria can be used. The expressions of AIC and BIC correspond to:

$$\text{AIC} = -2\ell(\widehat{\boldsymbol{\theta}}) + 2m,$$
$$\text{BIC} = -2\ell(\widehat{\boldsymbol{\theta}}) + m\log(n),$$

with $\ell(\widehat{\boldsymbol{\theta}})$ being the log-likelihood function evaluated at $\boldsymbol{\theta} = \widehat{\boldsymbol{\theta}}$ and $n, m$ being the sample size and number of model parameters, respectively. A smaller AIC or BIC indicates a better model. For more details on GD, AIC and BIC, see *Stasinopoulos & Rigby (2007)*.

## Density forecast

The density forecast (DF) technique aims to assess predictive performance outside the sample. This technique consists of dividing the original sample of data into a training set (2/3 of the first data) that is used to estimate the parameters, and then evaluate the performance outside the sample with the third rest of the data. That is, an out-of-sample test. These out-of-sample tests should be considered in the model validation process, see details in *Rojas (2016)*. A probability integral transform (PIT) is the cumulative probability

evaluated at the actual, realized value of the target variable. It measures the likelihood of observing a value less than the actual realized value, where the probability is measured by the DF. The PIT is uniform, independent and identically distributed if the density forecast is correctly specified.

### *Forecast with GSARIMA or GLARMA models*

Forecasts using GSARIMA or GLARMA models may be carried out in an analogous manner to GARCH models; see *Tsay (2009)*. Thus, based on GSARIMA or GLARMA models, and supposing that the forecast origin is $j = n$ and its horizon is $h$, we have the $h$-step ahead forecast is obtained from $y_{n+h}$, with initial prediction or fitted value $\hat{y}_n$ at the origin $n$ and forecast error $e_n(h) = y_{n+h} - \hat{y}_n(h)$, for $h \geq 1$.

## GSARIMA Forecast model for prevention of foodborne outbreaks by non-typhoidal salmonellosis

We adapted our methodological framework from *Maëlle, Dirk & Michael (2014)*. In this context, we considered an a-head time series based on fitted values of GSARIMA to alert infectious outbreaks, whose can be represented for multivariate forecast time series of counts. Denote the counts as $y_{it}$; $i = 1; \cdots; m; t = n+1; \cdots; n+h$, where $n+h$ is the length of the forecast time series, whose begin at time $n+1$ and $m$ is the number of entities, e.g., geographical regions, hospitals or age groups, being monitored. In the context of a forecast model for future disease outbreaks, it is essential to detect future changes in the process occurring at an unknown time $\tau$. As noted by *Sonesson & Bock (2003)*, this change can be a step increase of the counts of future cases or a more gradual change. Based on the possibility of such change, for each future time $t$ we want to differentiate between the two states in-control and out-of-control. At any timepoint $t_0 \geq n+1$ the available information -i.e., fitted values counts - is defined as $\hat{\mathbf{y}}_{t_0} = \{\hat{y}_t : t \leq t_0\}$. Detection is based on a statistic $m(\cdot)$ with a resulting alarm time $T_A = \min\{t_0 \geq 1 : m(\hat{\mathbf{y}}_{t_0}) > a\}$ where $a$ is a known threshold. The functions for outbreaks detection use fitted values to estimate $\mathbf{y}_{t_0}$, and compare it to the threshold $a$, above which, the current count can be considered as suspicious and, therefore, doomed as out-of-control. Then, based on *Farrington et al. (1996)* we designed a forecast model of outbreaks that uses the function of an algorithm called `algo.farrington` of survillance a package, in R software, see *Höhle & Riebler (2005)*. R is a non-commercial and open source software for statistical analyses and plotting, which can be obtained from http://www.r-project.org. We modified this function that summarize the *Farrington et al. (1996)* algorithm, to take a range of $h$-step ahead forecast values of the number of counts of infection reports. This $h$-step ahead forecast values are obtained from the multivariate time series GSARIMA forecast model show in 'Forecast with GSARIMA or GLARMA models'. For each time ahead, we use a GSARIMA simulated time series to set the number of counts in the same periods as a comparison. To obtain GSARIMA simulated time series we take command `garmsim` of `gsarima` package in R software, see *Briët, Amerasinghe & Vounatsou (2013)*. This command requires an autoregressive representation obtained by `arrep` function of same package. The estimated parameters from the time series of the observed data, necessary to define the autoregressive representation, can be estimated by `glarma` function of the same name package in R software, see *Dunsmuir (2015)*. Then, this

is compared to the forecast values number of counts. If the forecast value is above a specific quantile of the prediction interval given by simulated time series, then an alarm is raised, see Algorithm 1.

---

**Algorithm 1** Generation of alarm for prevention of outbreaks

---

1: Fit of the multivariate GSARIMA model from observed data and initial estimation of mean and overdispersion.
2: Generate simulated data of length to require ahead.
3: Forecast time point ahead of length to require with multivariate GSARIMA model.
4: Calculation of the weights omega (correction for past outbreaks).
5: Refitting of the model.
6: Revised estimation of overdispersion.
7: Rescaled model.
8: Omission of the trend, if it is not significant.
9: Repetition of the whole procedure.
10: Calculation of the threshold value.
11: Computation of exceedance score to generate a alarm.

---

# RESULTS

## Simulation study

The new methodology proposed in this paper is studied using a Montecarlo (MC) simulation study.

The GSARIMA forecast is simulated by using the autoregressive representation method, which is implemented in R, using a package named `gsarima`. This package contains methods for the generation of random numbers with univariate structure of time series, which is expandable to a multivariate response. This package has functions that allow the generation of generalized time series for a set of statistical counting distributions that belong to the exponential family. We focused on the negative binomial distribution (NBI), given its common use in this type of modeling, due to its properties of accumulation of counts.

We generated 5,000 scenarios of simulation, establishing different conditions given by a set of parameters in each scenario, in order to verify if the amount of alarms generated varies according to the configuration of the scenarios. For example, scenarios with a positive or negative trend must have values of $d > 0$ and / or $D > 0$, and positive or negative autoregressive coefficients, respectively. In turn, scenarios with coefficients of positive moving average reveal upward trends, and vice versa. To generate these 5,000 scenarios, we used the following indicators with the mentioned distribution, selected following *Briët, Amerasinghe & Vounatsou (2013)*:

## Statistical parameters of NBI statistical distribution
- dispersion parameter: {2,4}
- mean: {7,10}

## GSARIMA parameters
- autoregressive parameter: $p \sim U(-1,1)$
- moving average parameter: $q \sim U(-1,1)$
- seasonal autoregressive parameter: $P \sim U(-1,1)$
**Table 1  Classification for Monte Carlo study.**

| Type of parameter | Condition | Denomination | Condition | Denomination |
|---|---|---|---|---|
| autoregressive parameter | $p < 0$ | negative ar | $p > 0$ | positive ar |
| moving average parameter | $q < 0$ | negative ma | $q > 0$ | positive ma |
| seasonal autoregressive parameter | $P < 0$ | negative Sar | $P > 0$ | positive Sar |
| seasonal moving average parameter | $Q < 0$ | negative Sma | $Q > 0$ | positive Sma |
| integration of time series parameter | $d = \{1, 2\}$ | with integration | $d = 0$ | without integration |
| integration of seasonal time series parameter | $D = \{1, 2\}$ | with seasonal integration | $D = 0$ | without seasonal integration |
| dispersion parameter | 2 | Low dispersion | 4 | High dispersion |
| mean parameter | 7 | Low mean | 10 | High mean |

- seasonal moving average parameter: $Q \sim U(-1, 1)$
- integration of time series parameter: $d = \{2, 1, 0\}$
- integration of seasonal time series parameter: $D = \{2, 1, 0\}$
- covariate: $x \sim U(0, 1)$
- regressor coefficient: $\beta = 0.7$
- intercept: $\beta_0 = 1$

Using these parameters, we simulated 5,000 predicted GSARIMA time series of 156 weeks, each with a frequency of 52 weeks/years (total of 3 years). In each scenario, the generated outbreak alarms were recorded following the Farrington algorithm adapted to our methodology, see Algorithm 1. To verify the differences in the quantities of alarms, in the different generated scenarios, we constructed the classification shown in Table 1, according to the parameters of our Monte Carlo simulation study.

In order to show the different options that we can explore in our simulation study, we showed only three examples of simulated GSARIMA time series (ts) of length 156 week (3 years), using the approach mentioned in 'GSARIMA Forecast model for prevention of foodborne outbreaks by non-typhoidal salmonellosis'. The parameters of the simulated GSARIMA ts were: $ts1 = \{ p > 0, q > 0, P < 0, Q < 0, d = 0, D = 2 \}$, $ts2 = \{ p > 0, q > 0, P < 0, Q > 0, d = 0, D = 0 \}$, $ts3 = \{ p < 0, q > 0, P > 0, Q > 0, d = 1, D = 0 \}$. All these simulated GSARIMA ts were made considering high dispersion and high mean parameters, and can be examined in Fig. 1. On the abscissa axis, the time is shown in years divided weekly, where for example 2018.5 means the 26th week of the year 2018 (which marks the middle of the year 2018). In Fig. 2 the upperbound is shown as a dashed line (this band shows four-week aggregated data), while the alarms -timepoints where the upperbound has been exceeded- is shown as triangles. This date was obtained by the application of Algorithm 1 to generation of alarms for prevention of outbreaks in forecast GSARIMA ts1,ts2 and ts3.

We used the Kruskal-Wallis (KW) test to compare medians of indicators related to the grouped scenario denominations according to the denominations shown in Table 1. Results of comparative of mean/median, value of statistic KW and its $p$-value by grouped scenario denominations are showed in Table 2 .
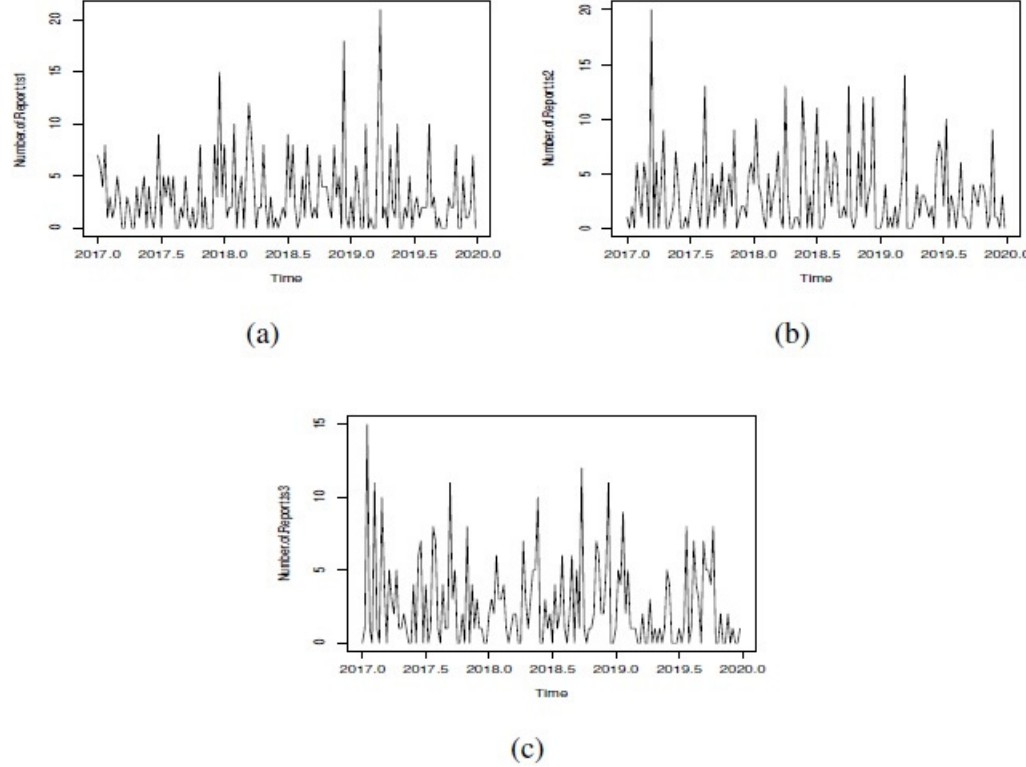

**Figure 1  Simulated GSARIMA: (A) ts1, (B) ts2 and (C) ts3.**

## Case study

In order to validate our model, we performed a case study based on data reports of *Salmonella enterica* serovar Enteriditis cases in Sydney, Australia (2014–2016). These data were weekly collected during three years (2014–2016) by the National Notifiable Diseases Surveillance System (NNDSS) from the Health Department of the Australian Government.

Figure 3 shows weekly time series of number of reports of *Salmonella enterica* serovar Enteriditis cases, whereas Fig. 4 presents (a) Autocorrelation function (ACF), (b) Partial Autocorrelation function (PACF) and (c) Negative binomial (NBII) Quantil-Quantil (QQ)-plots of the variable under study (weekly time series of number of reports of *Salmonella enterica* serovar Enteriditis cases), respectively. ACF and PACF show measures of the correlation between the observations of a time series separated by $k$ lag time units, in this case, $k = 1$ week. In our case, we have autocorrelation in several weeks of lag, whose are shown on the lines that exceed the confidence intervals shown by the scored lines. QQ plot indicates the comparison between theoretical quantiles (given by the proposed statistical distribution, NBII) and empirical data of the variables under study, whose all appear within the confidence intervals shown by the scored lines. Note that these figures show that the probability distribution of these variables are NBII but not independent and identically distributed. The adjusted forecast to deal with a zero inflation is covered by the

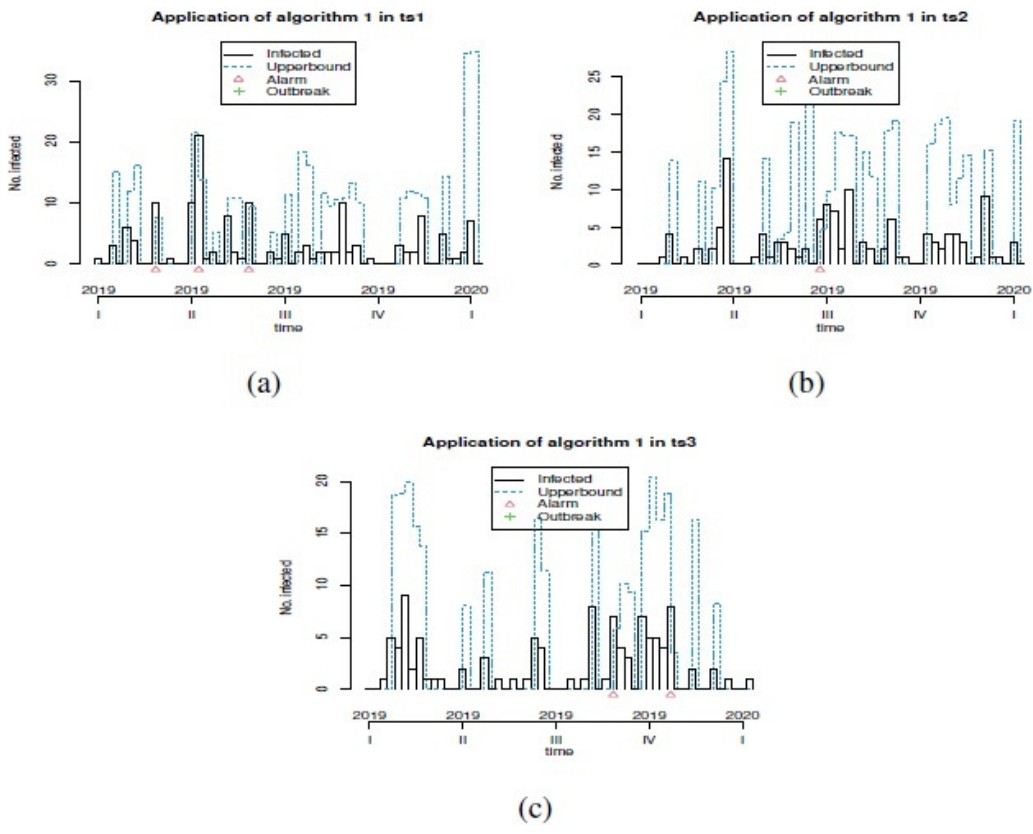

**Figure 2** **Plot of generation of alarms for prevention of outbreaks in forecast GSARIMA: (A) ts1, (B) ts2 and (C) ts3.**

**Table 2  Results of MC simulation study.**

| Type of parameter | Denomination | Mean/Median | Denomination | Mean/Median | KW | *p*-value |
|---|---|---|---|---|---|---|
| autoregressive parameter | negative ar | 2/1.73 | positive ar | 2/1.87 | 1.7158 | 0.1902 |
| moving average parameter | negative ma | 2/1.74 | positive ma | 2/1.85 | 9.9086 | 0.0016 |
| seasonal autoregressive parameter | negative Sar | 2/1.76 | positive Sar | 2/1.83 | 1.818 | 0.1775 |
| seasonal moving average parameter | negative Sma | 2/1.72 | positive Sma | 2/1.87 | 17.834 | 0.00002 |
| integration of time series parameter | with integration | 2/1.75 | without integration | 2/2.12 | 66.72 | 3.129e−16 |
| integration of seasonal time series parameter | with seasonal integration | 2/1.74 | without seasonal integration | 2/2.1 | 62.122 | 3.229e−15 |
| dispersion parameter | Low dispersion | 2/2.09 | High dispersion | 1/1.37 | 409.68 | <¡ 2.2e−16 |
| mean parameter | Low mean | 2/1.8 | High mean | 1/1.45 | 2.4248 | 0.1194 |

proposed statistical distribution, of a negative binomial type, which is capable of admitting data at zero, assigning to this value a probability distribution according to its historical and temporal frequency. To confirm that the original time series is stationary, we applied
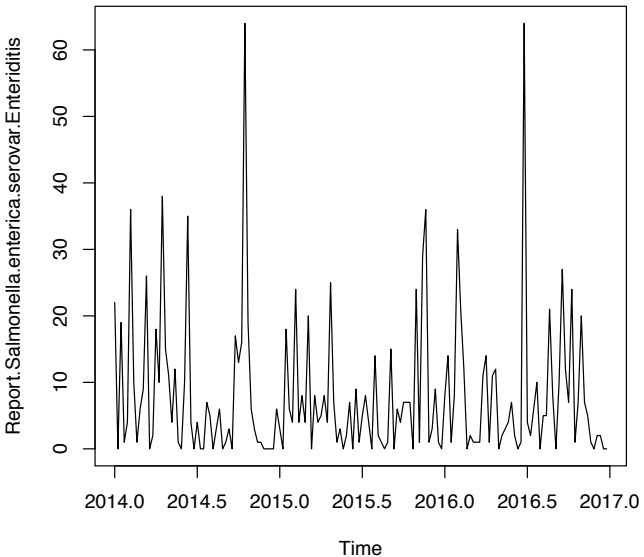

**Figure 3   Time series weekly of number of reports of *Salmonella enterica* serovar Enteriditis cases.**

Augmented Dickey-Fuller Test, obtaining a statistics Dickey-Fuller $= -4.0722$, Lag order $= 3$, $p$-value $= 0.0144$ for alternative hypothesis of time series with stationarity.

In this study we explored the relationship between the number of reports of *Salmonella enterica* serovar Enteriditis cases and different covariates obtained for the same time periods and city mentioned above. The consumption of undercooked eggs is known to be a risk factor for non-typhoid salmonellosis (*Martelli & Davies, 2012*) and infectious outbreaks caused by *Salmonella enterica* serovar Enteritidis have been widely reported worldwide (*Pijnacker et al., 2019*; *Jiang et al., 2020*; *Muvhali et al., 2017*; *Rizzo, 2006*). Additionally, environmental factors like temperature and humidity were included in this study. High temperatures have been previously correlated to an increased incidence of *Salmonella* infections (*Akil, Ahmad & Reddy, 2014*; *Kovats et al., 2004*), while humidity has been positively associated to *Salmonella* infections (*Kim et al., 2015*).

In this study, the covariates corresponded to the climatic variables of mean maximum temperature per week (°C/week), mean 3pm relative humidity (%/week), as well as the weekly demand for eggs (units/person*week). The climatological data were collected from the Bureau of Meteorology of the Australian Government, while egg consumption data were collected from the Euromonitor Agency. Figure 5 shows scatterplots between number of reports of *Salmonella enterica* serovar Enteriditis as response variable and the mentioned covariates. Note that, in general, the relationships between reports of *Salmonella enterica* serovar Enteriditis as response variable and the mentioned covariates are linear and positives. The covariates seemed to have a symmetric distribution.

We assumed a GSARIMA$(p, d, q)$ model, with $p = \{0, 1, 2\}$, $d = 0$ and $q = \{0, 1\}$ using a NBI statistical distribution for the number of reports de *Salmonella enterica* serovar Enteriditis cases and considered the covariates described with different link functions for
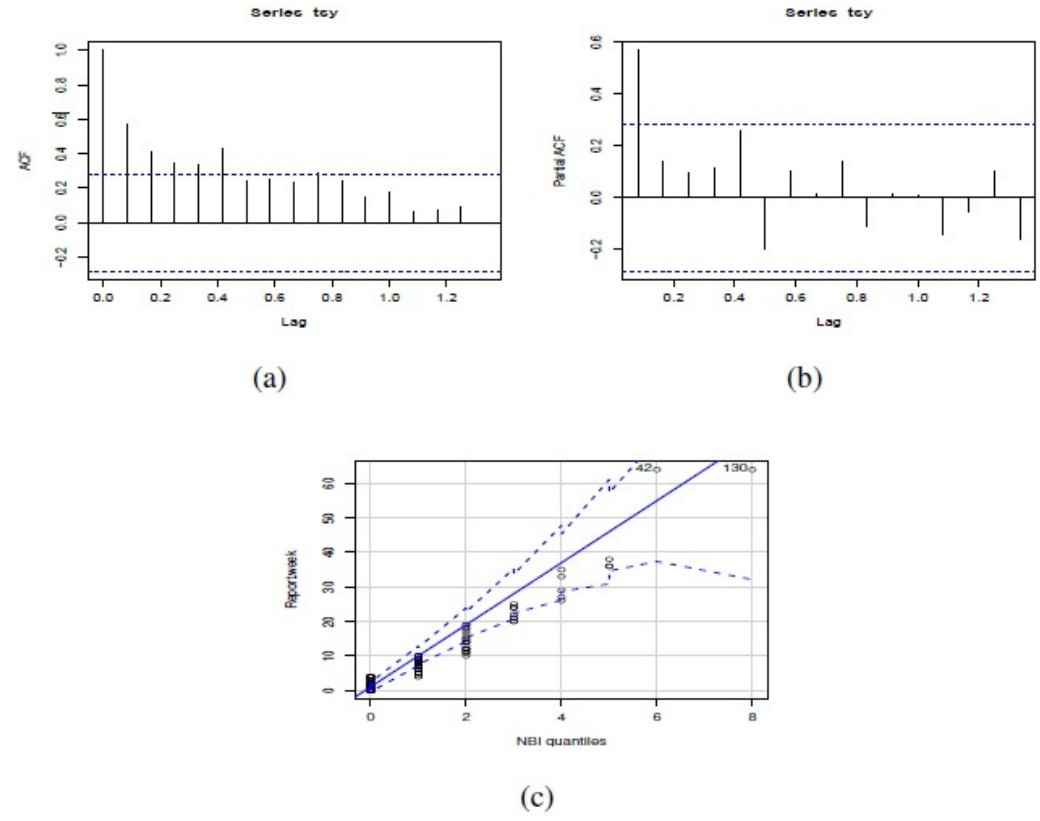

**Figure 4** (A) ACF, (B) PACF and (C) NBII QQ-plot of the number of actual reports of *Salmonella enterica* serovar Enteriditis cases of the dataset in study.

the mean response. In order to select the best GSARIMA model, we used AIC, BIC and GD, whose are reported on Table 3.

From Table 3, note that the smallest AIC, BIC and GD correspond to the GSARIMA(2.0,0) model with an identity link function, which is equivalent to a GLARMA model (2.0). In order to model the response variable of number of reports of *Salmonella enterica* serovar Enteriditis ($Y$), we propose the GLARMA model given by

$$E(Y_j) = \mu_j = \beta_0 + \beta_1 x_j + \phi_1 (y_{j-1} - \beta_0 - \beta_1 x_{j-1}) + \phi_2 (y_{j-2} - \beta_0 - \beta_1 x_{j-2}), \tag{8}$$

where $\beta_0$ and $\beta_1$ are the regression coefficients, $x_j$ is the value of the covariate vector $X = x_1, x_2, x_3$, where $x_1 =$ Mean Maximal Temperature (° C/week), $x_2 =$ Mean 3pm Relative Humidity (%/week), $x_3 =$ weekly demand for eggs (units/person*week) and $\phi_1, \phi_2$ are the autoregressive coefficients. We fit the GLARMA model by using the command `garmaFit`. The maximum likelihood estimates of the parameters of the model given in Eq. (8), with approximate estimated standard errors in parenthesis, are: $\widehat{\beta}_0 = -15.28(0.86)$, $\widehat{\beta}_1 = \{\widehat{\beta}_{x_1} = 0.25(0.019), \widehat{\beta}_{x_2} = 0.19(0.02), \widehat{\beta}_{x_3} = 0.95(0.58)\}$, $\widehat{\phi}_1 = -0.021(0.045)$, $\widehat{\phi}_2 = -0.014(0.045)$ and $(\widehat{\text{Var}}(Y_t|\mathbf{H}_t))^{1/2} = (\widehat{\varphi}(\mu_t))^{1/2} = 0.041(0.016)$. All coefficients are significant at 10%. The coefficient of determination ($R^2$) of this model is 0.88. This indicates that the variables considered in the model ($y_{t-1}, y_{t-2}, x_1, x_2, x_3$) explain 88%

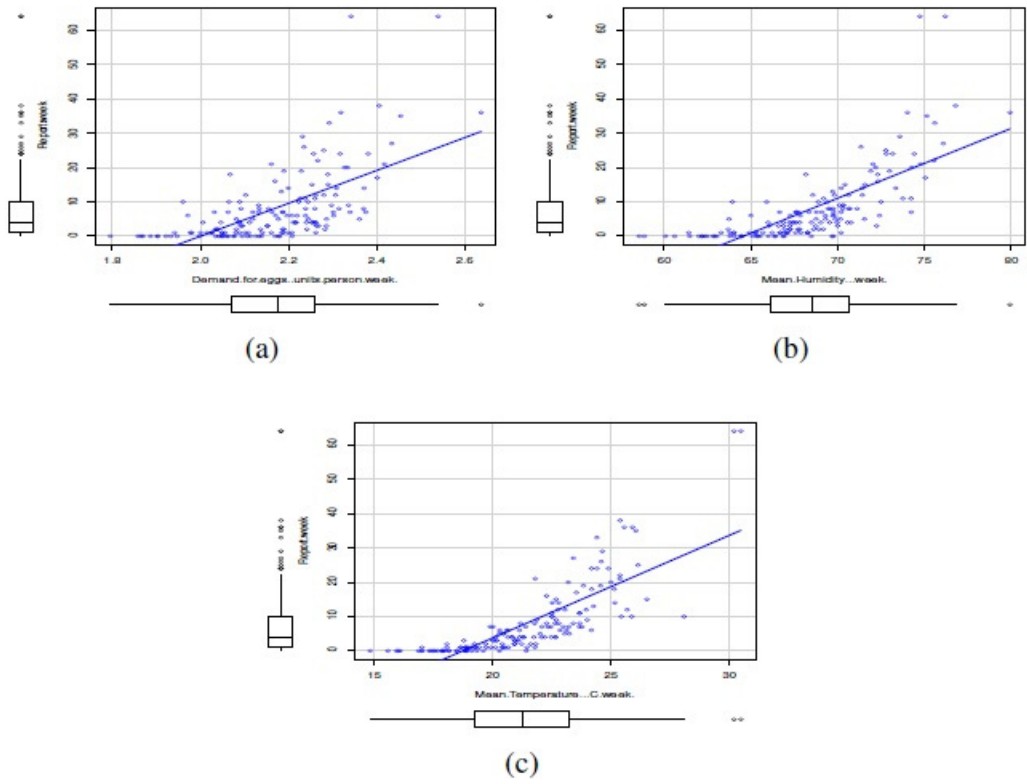

**Figure 5** Scatterplots between number of reports of *Salmonella enterica* serovar Enteriditis as response variable and covariates: (A) weekly demand for eggs (units/person*week), (B) mean 3 pm relative humidity (%/week), and (C) mean maximum temperature per week (°C/week).

of the variations of the response variable ($y$). In this model not all explanatory variables contribute the same in determining the variation of the response variable. For example, extracting the variable $x_3$ from the model $R^2$ remains almost unchanged (0.87), while extracting $x_3$ and $x_2$, $R^2$ decreases to 0.73, leaving only the lags $y_{t-1}, y_{t-2}$ explaining the response variable $y$, $R^2$ decreases to just 0.09. Then, prediction model can be expressed as

$$\widehat{\mu}_{j+1} = -15.28 + 0.25x_{1_{j+1}} + 0.19x_{2_{j+1}} + 0.95x_{3_{j+1}}$$
$$-0.021(\widehat{y}_j - (-15.28 + 0.25x_{1_j} + 0.19x_{2_j} + 0.95x_{3_j})) +$$
$$0.041(\widehat{y}_{j-1} - (-15.28 + 0.25x_{1_{j-1}} + 0.19x_{2_{j-1}} + 0.95x_{3_{j-1}})).$$

To confirm the correct fit of this proposed GLARMA model, six plots were examined in Fig. 6: (a) Observed time series related to fixed effect of GLM estimation or GLARMA estimation;, (b) Disposition of Pearson Residuals in the time; (c) Histogram of Uniform PIT, (d) Histogram of Randomized Residuals normalized (randomized for a discrete response distribution); (e) Quantile-Quantile (QQ) Plots for randomized residuals of a fitted GLARMA object; and (f) Plot of the ACF of the residuals. These results indicated that is possible to apply a GLARMA model to original data, considering the covariates and logarithmic link function to the mean of response variable in each time of its realization. The selection of the best GLARMA model order is according Akaike criteria. Addiotonally,

**Table 3  Criterion and GD for different GSARIMA models with actual data of the number of reports of *Salmonella enterica* serovar Enteriditis cases.**

| Information criterion | GSARIMA(0,0,0) | | GSARIMA(1,0,0) | | GSARIMA(0,0,1) | | GSARIMA(1,0,1) | | GSARIMA(2,0,0) | | GSARIMA(2,0,1) | |
|---|---|---|---|---|---|---|---|---|---|---|---|---|
| | Identity | Log | Identity | Log | Identity | Log | Identity | Log | Identity | Log | Identity | Log |
| AIC | 691.56 | 693.61 | 665.67 | 657.32 | 657.12 | 659.29 | 653.22 | 651.06 | 646.73 | 869.17 | 644.65 | 867.33 |
| BIC | 694.34 | 697.23 | 676.98 | 664.12 | 669.44 | 668.64 | 656.84 | 660.42 | 657.96 | 880.40 | 656.43 | 878.14 |
| GD | 683.76 | 683.22 | 657.94 | 650.42 | 648.55 | 648.29 | 634.93 | 641.06 | 634.73 | 857.17 | 633.12 | 856.72 |

we also checked Box–Pierce test type Ljung to corroborate aleatory disposition to 2 lags with statistic $\chi^2 = 1.18$, degree of freedom (df) = 2, and $p$-value = 0.582, and normal distributions of the randomized residuals of the model, with Shapiro–Wilk normality test obtained statistic $W = 0.9874$, $p$-value = 0.9321. The forecast PDF serves to simulate future scenarios of number of reports of *Salmonella enterica* serovar Enteriditis cases.

Following Algorithm 1 we can apply our model for prevention of future foodborne non-typhoidal salmonellosis outbreaks considered the above mentioned forecast covariates projected for the next year. These weeks results are shown in Fig. 7. Our model makes it possible to predict 3 alarms in the third quarter of next year, given the expected weather and egg consumption conditions. Remember that upperbound showed as a dashed line shows four-week aggregated data. In this case, the model does not forecast outbreaks.

# DISCUSSION

In this work, we developed a forecast model for the prevention of foodborne outbreaks due to non-typhoidal salmonellosis, which is useful for alerting future infectious outbreaks related to the intake of foods considered of risk.

We validated our alert model for infectious outbreaks related to food consumption and weather conditions, based on real time series data of reports of *Salmonella enterica* serovar Enteriditis infections in Australia, which uses food and weather conditions as predictive variables. Through a very flexible statistical treatment granted by GSARIMA, which uses covariates and can adapt to a varied class of statistical counting distributions that can have different degrees of asymmetry as well as temporal effects of seasonality, it was possible to predict the adjusted value with high precision.

We analyzed the conceptual antecedents and the theoretical foundations that lead to the processes of our research modeling, which resulted useful for the implementation and applications of the science of administration in the field of epidemiology. Through this model, we intend to contribute with a useful tool for public health decision taking, that can accurately alert foodborne outbreaks of non-typhoidal salmonellosis. The ability to predict outbreaks of our model depends on the interaction of recognized hazards and associated conditions. Therefore, for the evaluation of the appearance of new food safety risk factors, it is necessary to study whether or not these risks are associated with outbreaks of infectious diseases, and how much they contribute to them. In this context, an application of our model could be to identify when the known or new factors can be useful or fail to predict

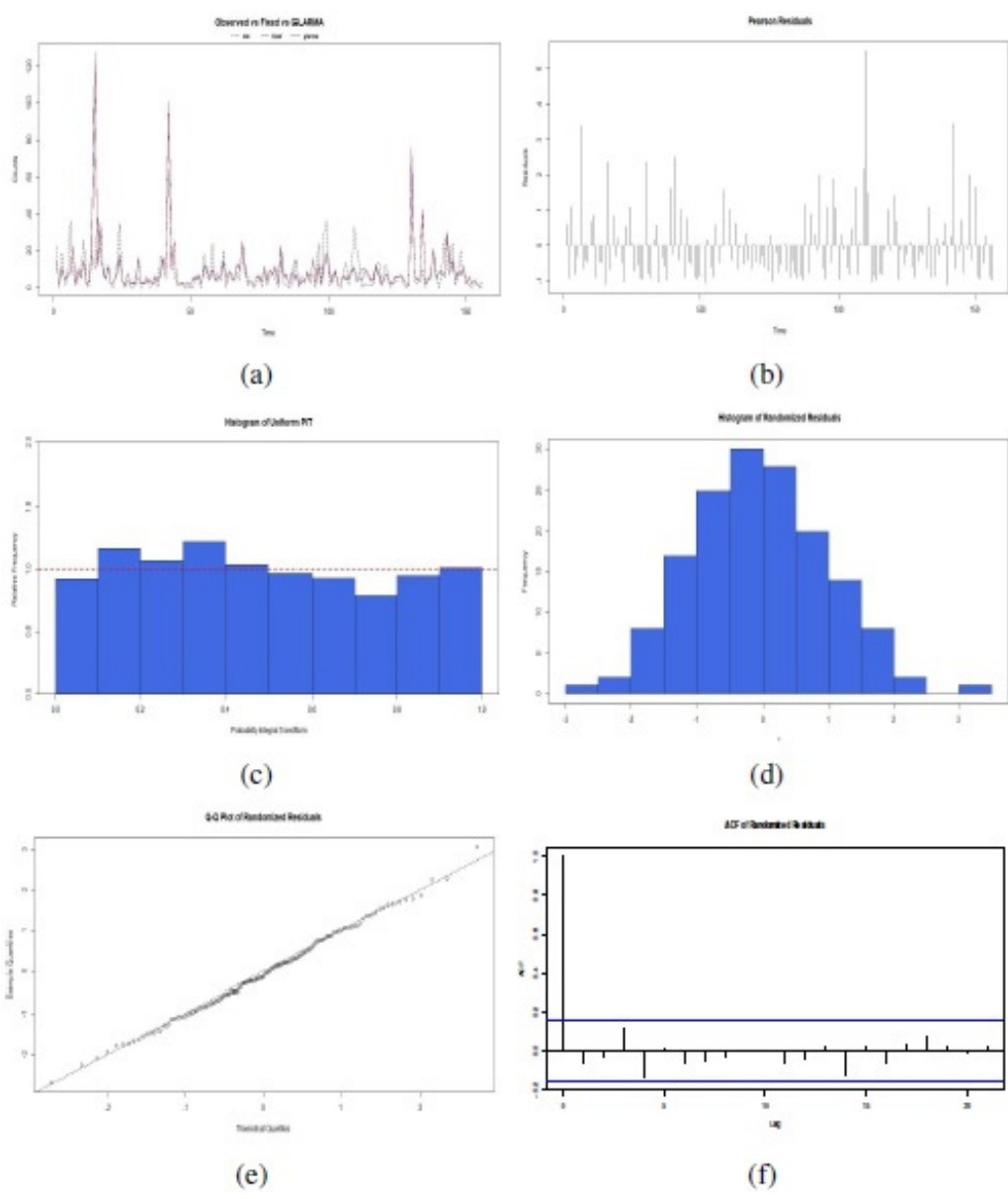

**Figure 6** Confirmatory plot analysis of the fit of the proposed GLARMA model for the response variable in the data set of the illustrative study: (A) Observed time series related to fixed effect of GLM estimation or GLARMA estimation; (B) Pearson Residuals; (C) Histogram of Uniform PIT; (D) Histogram of Randomized Residuals normalized, (E) QQ Plots for randomized residuals of a fitted GLARMA ; and (F) Plot of the ACF of the residuals (source: authors).

the appearance of diseases. Failure to predict a known factor could be a signal for the emergence of a new food safety hazard.

The main limitations of our model are related to the homoscedasticity assumption of the infection report count data. Therefore, this limitation leads to a possible future topic of

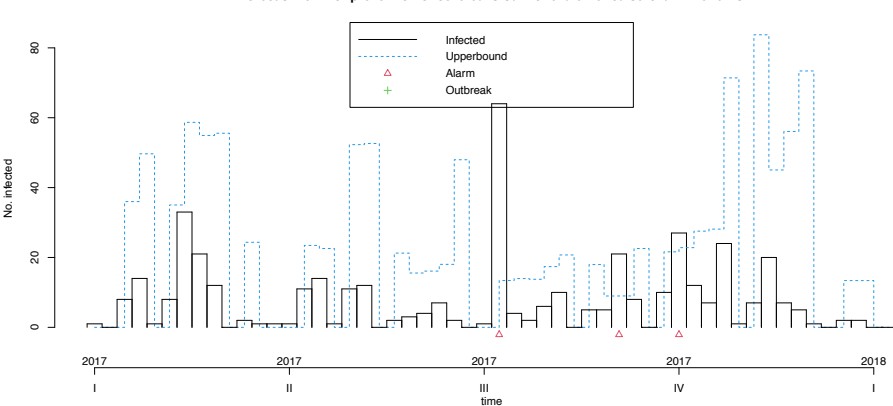

**Figure 7** Model for the prevention of foodborne outbreaks produced by *Salmonella enterica* serovar Enteriditis to the illustrative response variable in the dataset in study (source: authors).

reserach. Apparently, this limitation could be overcome by using models of multivariate-type generalized autoregressive conditional heteroscedasticity (GARCH), whose have some attractive properties, like a greater weight on the most recent observations, but also inconveniences like an arbitrary disintegration factor that introduces subjectivity in the estimation, see *Thamanukornsri & Tiensuwan (2018)*.

## CONCLUSION

We propose a forecast model for non-typhoidal salmonellosis outbreaks, a foodborne illness, which in the case of *Salmonella enterica* serovar Enteritidis is related to the consumption of eggs and climatic factors of humidity and ambient temperature.

The proposed methodology uses the modeling of infection reports by mean of GSARIMA model, which allows the use of predictive covariates. Additionally, this model can raise an alarm when a high probability of a foodborne infectious outbreak is detected, which can be useful in the surveillance and management of health care.

### Funding

This research was funded by the Centro de Micro-Bioinnovación, Universidad de Valparaíso, Chile (DIUV-CIDI 4/2016), and grant "Fondecyt de Iniciación 1119004" (Fernando Rojas) from the National Agency of Research and Development of Chile. There was no additional external funding received for this study. The funders had no role in study design, data collection and analysis, decision to publish, or preparation of the manuscript.

### Grant Disclosures

The following grant information was disclosed by the authors:

Centro de Micro-Bioinnovación, Universidad de Valparaíso, Chile: DIUV-CIDI 4/2016.

Fondecyt de Iniciación: 1119004.
National Agency of Research and Development of Chile.

## Competing Interests

The authors declare there are no competing interests.

## Author Contributions

- Fernando Rojas conceived and designed the experiments, performed the experiments, analyzed the data, prepared figures and/or tables, authored or reviewed drafts of the paper, and approved the final draft.
- Claudia Ibacache-Quiroga conceived and designed the experiments, performed the experiments, authored or reviewed drafts of the paper, and approved the final draft.

## Data Availability

Raw data is available as a Supplemental File.

## Supplemental Information

Supplemental information for this article can be found online at http://dx.doi.org/10.7717/peerj.10009#supplemental-information.

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
