# Peer review of "A forecast model for prevention of foodborne outbreaks of non-typhoidal salmonellosis"

_PeerJ, doi:10.7717/peerj.10009_

## Round 0.1 · original submission · Major Revisions

Your manuscript has been reviewed and requires modifications prior to making a decision. The comments of the reviewers are included at the bottom of this letter. Reviewers indicated that methods and results sections should be substantially improved. Reviewers also recommended extensive English editing. I agree with the evaluation and I would, therefore, request for the manuscript to be revised accordingly.

·

Basic reporting

No comment.

Experimental design

No comment.

Validity of the findings

No comment.

Reviewer 2 ·

Basic reporting

The paper is difficult to read due to the dense technical language and equations used to describe the models.

The figures a small. They are not well labeled.

Experimental design

I am not a statistician, and cannot comment on the extensive discussion of the development of the model.

Validity of the findings

The source of the data are described, but the data themselves are not. For example, is Salmonella enteritidis intended to refer to the serotype enteritidis, or to Salmonella enterica, generally? If the first, why is pork demand being used to predict the occurrence of a serotype associated primarily with chickens? If the second, the data need to be properly described.

How is predicting the occurrence of an outbreak based on generic distribution of a commodity useful for prevention?

Additional comments

The authors have developed a statistical model to help predict the occurrence of non-typhoidal Salmonella outbreaks. Much of the paper lays out the theoretical basis and historical development for the modeling approach. This is rich in technical terms and equations that are frankly beyond my expertise. The paper needs to be reviewed by a statistician to vet the language, equations and approach.
My own expertise as an epidemiologist relates to the identification, use and interpretation of the data used to validate the model. The authors present a case study for predicting outbreaks of Salmonella in Sydney, Australia based on estimates of mean maximum temperature, mean 3 pm relative humidity, and weekly demand for pork. They apply their algorithms to the data and forecast the occurrence of Salmonella based on forecasts of the occurrence of the covariates.
There is very little information presented about the data used in these models. The first question relates to the Salmonella counts. The title of the paper discusses nontyphoidal Salmonella, which encompasses a wide range of different Salmonella serotypes and likely reservoir sources. However, the paper discusses obtaining counts of Salmonella enteritidis. Salmonella enterica serotype Enteritidis is a specific serotype of Salmonella that has been primarily associated with chickens. The organism colonizes chickens asymptomatically and can infect the hen’s ovaries. This led to global problems with S. Enteritidis transmission as a result of egg production. More recently, S. Enteritidis has become a recognized problem with chicken meat production. While the serotype is not biologically restricted to chickens, it is not clear why its occurrence would be predicted by consumption of pork. If the authors are using the “Salmonella enteritidis” to refer to Salmonella enterica, generally, they need to revise their description of the data. With the three years of data on case counts obtained, how many outbreak-associated cases were included? Distinguishing between sporadic cases and outbreak-associated cases could influence the development and use of the data. In the US, 5-10% of Salmonella cases are associated with an outbreak. Large outbreaks embedded in the data could affect the modeling, and any adjustments made to account for the possible presence of outbreaks in the data should be described.
The authors should also discuss the rationale for the use of temperature and humidity in the models. Temperature is certainly known to affect the growth of Salmonella. However, what data support the affect of humidity? Certainly, temperature and humidity are seasonally associated environmental factors, but it is not clear what the independent contribution of humidity may be.
The final point I would raise is on the usefulness of predicting the occurrence of outbreaks. It is true that Salmonella has a strong seasonality with increased occurrence of cases and outbreak potential during the summer months. However, outbreaks occur within defined settings, or associated with specific food exposures. A predicted increase in pork consumption during a hot, humid stretch, could lead to public health warnings. Is this what the authors envision? Every year in the US such warnings are issued with respect to the advent of outdoor cooking of hamburger to prevent E.coli O157:H7 infections. This type of general advice does not require the use of elegant models. Beyond this, what specificity of prediction is possible. On a global basis, the incidence of Salmonella infections appears to be slowly increasing. However, one of the reasons for this is the broad host range of nontyphoidal Salmonella. Thus, focusing on a single commodity, would not address the actual population exposure risk. If the models could be broadened to accommodate the full range of Salmonella attribution, it would be interesting. Perhaps this initial model could serve as a test of concept for a broader approach.
In terms of the construction of the manuscript, the technical language and volume of equations makes reading the paper difficult. There are numerous figures present that are reproduced at such a small scale as to make them difficult to interpret. Also, the labels on the figures do not adequately describe the data. For example, in figure 5, it is not clear what the various scatterplots are showing.

·

Basic reporting

The manuscript is overall well presented, however the English language in most parts is grammatically incorrect, unclear and very difficult to interpret. For instance, l. 11-15 (convoluted and overlong sentence with grammatical errors), l. 18-19 (‘health public’ is a wrong term) – and generally throughout the manuscript. I suggest that the authors pay very close attention to writing clear, concise sentences – particularly as this is quite a technical manuscript which otherwise can become too hard to read.
The figures presented need more information:
• What exactly is presented in Figure 1? Are these predicted Salmonella cases? What is the time line (it does not correlate with either 156 weeks or 3 years).
• The axes on Figure 2 are hard to read.
• Figure 4 needs more elaboration: again what is the variable under study? What are the three different lines in part (c) – they are hard to read.
• Figure 5 lacks a clear explanation; without reading the main text it is almost impossible to determine what this figure depicts and even then it is hard to interpret. Consider presenting these data in a different way.
• Figure 7 is possibly the most interesting figure in the manuscript but again hard to interpret. Does ‘upperbound’ indicate the upper threshold? What does ‘infected’ mean – predicted or observed numbers? I can’t see any green crosses in this figure although the legend indicates them – and what do they mean, an observed outbreak? Please also indicate that the forecast is per month (and provide a scientific rationale for this choice).

Experimental design

Reading through the manuscript, it is hard to understand what parameters specifically relating to Salmonella the forecast model is based on. Or does the model estimate general parameters for infectious/foodborne diseases which are then used for the forecasting? Please describe exactly the background knowledge, data (and not just the mathematical assumptions) that went into developing the model. For an epidemiologist and any public health official working with disease surveillance, this is crucial knowledge.
In the case study, the authors use relative humidity, temperature and weekly demand for pork to predict the future number of Salmonella cases. Where is the scientific basis for using exactly these variables? Salmonellosis is often a highly seasonal disease and so any variable that fluctuates with season could technically be used in a forecasting model, but without much epidemiological sense. Also, the forecast does not account for either serovariant, subtype or geographical variation which is a serious omission as almost all Salmonellosis outbreaks are firstly determined by serovar or (genetic)subtype and often geographically limited. The authors state that their model covers Salmonella enteritidis infections, however, the raw dataset includes many non-enteritidis cases. Were these excluded from the forecast? And what is the purpose of focusing solely on enteritidis when Salmonella Typhimurium is the most common serovar in Australia?
Lastly, I am concerned about the forecasting in general which is not well described - how long ahead does the model validly forecast the number of cases? Is it on a weekly basis or monthly basis as indicated in Figure 7? What is the scientific rationale behind this selection? How does the forecast deal with weeks/months where zero cases are observed, and was the forecasting adjusted to deal with a zero inflation?

Validity of the findings

Please refer to my comments above.

Additional comments

The authors present a potentially useful tool for predicting outbreaks of a common foodborne disease. However, the manuscript lacks important information which is needed for a proper epidemiological evaluation of the usefulness of the predictions. An infectious disease epidemiologist would need to see more disease-specific knowledge used in such a study in order for the results to be of public health importance. In general, the manuscript is highly technical and suitable for a specialized audience, and I think that the public health relevance is not particularly evident as it is presented now. The English language concerns also poses a hindrance for fully understanding the methods employed.

---

## Round 0.2 · Major Revisions

The authors addressed the reviewers' concerns and substantially improved the content of MS. Thanks for your detailed revision. However, Reviewer 2 still has concerns. The comments of the reviewer are included at the bottom of this letter. I agree with the evaluation and I would, therefore, request for the manuscript to be revised accordingly. The manuscript also needs extensive English editing because there are several typos and grammatical errors. In addition to this, please update the last two sentences of the “Acknowledgements” section.

Reviewer 2 ·

Basic reporting

There are numerous small grammatical errors throughout the manuscript. For example, lines 12,13 "for the an early alert...pathogen, whose are related...", line 18 "is a foodborne illness considered is a..".

Experimental design

As previously noted, the paper is dense with technical biostatistical language I am not competent to evaluate. The authors do not really address the issue of why better prediction is needed to improve source control (lines 23-24). There is another issue in that the ability of the model to predict outbreaks is dependent on the interplay of recognized hazards and associated conditions. It does not fundamentally allow for assessment of the emergence of novel food safety hazards. One application of such a model could be to identify when the known effects fail to predict the occurrence of disease, which could be a flag for the emergence of a new food safety hazard. Some consideration of this idea is warranted.

Validity of the findings

In the previous version of this paper, the authors included consumption of pork as a predictor. It was pointed out that pork is not a recognized vehicle for Salmonella Enteritidis. In this version of the paper, the authors have included consumption of eggs as a predictor. This is good. However, in comparing the two versions of the paper, it appears that Figure 7 is identical. The labels have been changed but the underlying figure and the three alarms generated appear to be the same. How is it that changing a major input, such as consumption of eggs for pork, has no impact on the predictive value of the model?

---

## Round 0.3 · accepted · Accept

The authors addressed the reviewers' concerns and substantially improved the content of MS. So, based on my own assessment as an academic editor, no further revisions are required and the MS can be accepted in its current form.

Reviewer 2 ·

Basic reporting

OK, technical writing is dense.

Experimental design

OK.

Validity of the findings

OK, but the model seems insensitive to the inclusion of specific exposure variables.

Additional comments

The authors have made suitable modifications to prior comments, and appropriately qualified the use of their predictive model.